# Longitudinal and Cross-Sectional Evaluation of Two Commercial Swine Breeding Herds to Characterize Neutralizing Antibody Levels following Porcine Epidemic Diarrhea Virus Outbreaks

**DOI:** 10.3390/v16030324

**Published:** 2024-02-21

**Authors:** Justin Brown, Kristin Skoland, Heather Kittrell, Josh Ellingson, Paul Thomas, Chelsea Ruston, David Baum, Locke Karriker

**Affiliations:** 1Swine Medicine Education Center, College of Veterinary Medicine, Iowa State University, Ames, IA 50011, USA; 2AMVC Management Services, Audubon, IA 50025, USA; 3Department of Veterinary Diagnostic and Production Animal Medicine, College of Veterinary Medicine, Iowa State University, Ames, IA 50011, USA

**Keywords:** porcine epidemic diarrhea virus, neutralizing antibody

## Abstract

Neutralizing antibodies to Porcine Epidemic Diarrhea Virus (PEDV) can be detected by 3 weeks post-infection and remain detectable through at least 24 weeks post-infection. The objective of this study was to evaluate the levels of neutralizing antibodies in sow and piglet serum and sow milk to determine the duration of neutralizing antibodies following PEDV outbreaks. Two farms were selected for the study following outbreaks of PEDV. Monthly, cohorts of sows were sampled and followed through two farrowings. Following each farrowing, samples from piglets and milk were collected. Samples were evaluated for PEDV-neutralizing antibodies by a high-throughput fluorescent neutralization assay. Although neutralizing antibodies to PEDV can be detected throughout 15 months post-outbreak, a decrease in circulating neutralizing antibody levels is noted in farms beginning at six months post-outbreak. With decreasing levels, farms may become more vulnerable to PEDV outbreaks, and practitioners can focus on this time window to implement intervention strategies.

## 1. Introduction

Porcine epidemic diarrhea virus (PEDV) is an *Alphacoronavirus* that has become endemic in the Americas following its emergence in the United States in 2013 [1,2,3]. PEDV continues to cause issues in the US Swine herd. Data from the Swine Disease Reporting System showed a cyclical pattern of positive cases, with the percentage of positive cases ranging from 6.2–16.2% from 2018 to 2023 [4]. Serology testing is a diagnostic tool to detect previously exposed animals and to determine if those animals may be protected from clinical disease. Following an outbreak of PEDV within a breeding herd, neutralizing antibodies can be detected in milk and serum samples within 3 weeks after the exposure and remain positive through 24 weeks post-exposure, as described by Clement et al. [5]. Poonsuk et al. [6] demonstrated that antibody-positive groups infected with PEDV returned to normal body temperature faster and demonstrated a higher survivability rate compared to negative antibody cohorts. Similar findings have been reported for other swine diseases. Lopez et al. [7] showed that the presence of a 1:8 titer of porcine reproductive and respiratory syndrome virus—neutralizing antibody in serum protected pigs against viremia but not peripheral tissue seeding and transmission of the virus. Yang et al. [8] demonstrated that high levels of Senecavirus A (SVA)—specific neutralizing antibodies could provide protection against SVA infection. These data demonstrate that neutralizing antibodies in serum may offer protection in piglets. Therefore, the objective of this study was to evaluate the neutralizing antibody levels in sow and piglet serum and sow milk to better define the duration of neutralizing antibodies within farms following an outbreak of PEDV.

## 2. Materials and Methods

### 2.1. Herd Selection

Two commercial breeding herds located in northwest Iowa, with recent outbreaks of PEDV in the spring of 2018, were selected for enrollment in the study following clinical outbreaks of PEDV. Farm 1 was comprised of 2400 females, and Farm 2 was comprised of 1500 females. Farm 1 was a continuous farrowing system, while Farm 2 used batch farrowing. At the time of enrollment, the herds were PEDV stable and no longer experienced losses or clinical disease. Sample collection commenced in April for Farm 1, and PEDV was detected in late December. Sample collection commenced in May for Farm 2, and PEDV was detected in April.

### 2.2. Animal Enrollment

Following enrollment of the farm, each month, 10 gilts and 10 sows (*n* = 20 per month) were enrolled as a cohort into the study. Each cohort was enrolled one month prior to farrowing and then followed through two consecutive farrowings. For enrollment into the study, sow and gilts were selected within a breed week by using the randomization function in Excel to select stalls where females were housed. For Farm 1, a total of 120 females were enrolled, and for Farm 2, a total of 180 were enrolled in the study.

### 2.3. Sample Collection

Following enrollment into the study, a blood sample was collected from each female via jugular venipuncture using a 16-gauge X 3-inch needle attached to a 12-milliliter syringe. The blood sample was then transferred to a serum separator tube and placed on ice for transportation to the laboratory. Blood tubes were centrifuged at 3600× *g* for 8 min to separate the serum. The serum was then transferred to 2.5 mL cryovials and stored at −80 °C until sample analysis. Samples were stored in duplicate. Approximately two weeks post-farrowing, three piglets were conveniently selected from the litter, and a blood sample was obtained via jugular venipuncture using a 20-gauge, 1-inch needle attached to a 12 mL syringe. The blood was then transferred to a serum separator tube and placed on ice for transportation to the laboratory. At 14 days post-farrowing, a blood sample was again collected from the sow as described above, and a milk sample was obtained by hand stripping each teat to collect a minimum of 5 mL. Post-farrowing samples were transported to the laboratory for processing. Serum samples were spun and stored as described previously. Milk samples were allocated to 2.5 mL cryovials in duplicate. Samples were stored at −80 °C until sample analysis.

Following the first farrowing, a blood sample was collected from each enrolled female once per month by the method described above. After her second farrowing, blood and milk samples and blood samples from three piglets were also collected as described above. Samples were processed and stored at −80 °C until sample analysis. Figure 1 illustrates the sample collection points for females in one cohort and which samples were collected.

### 2.4. Sample Analysis: High Throughput Fluorescent Neutralization (HTNT) Procedure

The HTNT was performed by the Iowa State University Veterinary Diagnostic Laboratory (Ames, IA, USA) standard operating procedure 9.5324 Version 2 [9]. Serum samples were thawed and heat-inactivated in a water bath at 56 °C for 30 min. Virus stock was serial diluted 10-fold in virus inoculation medium (DMEM with 2 µg/mL Trypsin-TPCK), transferred to Vero 81 cells (ATTC^®^ CCL-81^™^), and incubated at 37 °C with 5% CO_2_ for 5 days. The last dilution in which the cytopathic effect was observed was used to determine the TCID50 using the Spearman and Karber method [10].

One-hundred microliters of cell solution (Cell Propagation Medium, Tyrpsin-EDTA, and Vero 81 cells) was diluted with 100 µL of cell propagation medium (DMEM with 10% FBS and 1% penicillin-streptomycin) to make a 1:1 dilution. A hemocytometer was used to count and calculate the dilution to make 5 × 10^4^ cell/well and added to each well of a clear bottom black 96-well plate (Coring 96-well CellBind microplate, Sigma) and incubated for 48 h at 37 °C with 5% CO_2_ to make the cells confluent (100% confluent). Each plate had positive control, negative control, and virus control wells. Five microliters of heat-inactivated controls and samples were placed in duplicate wells to make 1:40 dilution by first 1:20 with adding 95 µL of virus inoculation medium, then 100 µL of 1:10 diluted PEDV virus (3.16 × 10^5^ TCID50/mL). The serum/virus mixture was incubated for 1 h at 37 °C with 5% CO_2_. All cell wells of the test plate were washed three times with 150 µL of cell washing medium (DMEM with 1% penicillin-streptomycin), and 150 µL of serum/virus mixture and virus controls were transferred to the test plate and incubated for 90 min at 37 °C. The serum/virus mixture was discarded, and plates were washed once with a cell washing medium, as described above, followed once with a virus inoculation medium. Virus inoculation medium (150 µL) was added to each well and incubated for 24 h at 37 °C with 5% CO_2_. The virus inoculation medium was discarded, and the plate was washed with PBS pH 7.4. Cells were fixed by adding 4 °C and 80% acetone to each well and incubated at room temperature for 15 min. Acetone was removed, and the plate was air dried for 30 min. Each well was rinsed with PBS pH 7.4. Next, 50 µL of FITC-conjugated PEDV NP monoclonal antibody (SD6-29 clone, Medgene Labs) diluted 1:100 in PBS pH 7.4 was = added to each well and incubated for 60 min at 37 °C. The plate was washed four times with PBS pH 7.4, leaving the last wash in the plate for 5 min, then discarded and replaced with 100 µL of PBS pH 7.4. Plates were then read with SpectraMax i3x using SoftMax Pro 6.5 (Molecular Devices, San Jose, CA, USA) using 30 ms exposure time and 20 um focal adjustments for 541 wavelengths. The total fluorescence reduction percentage (%FR) was calculated as 100 − ([Average sample Total Fluorescence Intensity/Average Negative Control Total Fluorescence Intensity]) × 100) (Sarmento et al., 2020). A higher %FR indicates higher neutralizing antibody concentrations within the samples. Samples with a FR ≥ 85% were classified as positive for neutralizing antibodies [2].

### 2.5. Statistical Analysis

All statistical analyses and descriptive statistics were evaluated using RStudio [11] statistical software (version 4.0.2). Cohorts were compared within each farm by mixed linear regression adjusted for heteroscedasticity. Pairwise comparisons of cohorts by month were evaluated using a linear mixed model, corrected for heteroscedasticity and Tukey’s adjustment. Relationships of antibody levels in sow serum, piglet serum, and milk were evaluated using Pearson’s product-moment correlation to determine the linear association of the variables by producing a sample correlation coefficient, r, as a representative of the population correlation coefficient, ρ. *p* ≤ 0.05 were considered statistically significant.

## 3. Results

The number of females sampled by month post-outbreak is shown in Figure 2. Farm 1 had six cohorts enrolled, and Farm 2 had nine cohorts enrolled.

Figure 3 and Figure 4 display the range of serum-neutralizing antibody levels by month post-outbreak for each cohort for Farm 1 and Farm 2, respectively. For Farm 1, there is a statistically significant difference in the mean level of serum-neutralizing antibody detected in months 9–12 compared to months 4–9 post-PEDV outbreak (Figure 3). As each cohort is enrolled in the study, there is an increase in variability of detected antibody levels, which is shown as a wider range of antibody levels by month. As time post-outbreak progresses for each farm, cohorts eventually start to fall out of the study, leading to less individual animal variation in the data and fewer potential outliers by month. For Farm 2 (Figure 4), there is a visual decrease in mean serum-neutralizing antibody levels beginning at month 4 post-outbreak, but a statistical difference is not observed until month 6 post-outbreak.

Figure 5, Figure 6, Figure 7, Figure 8, Figure 9 and Figure 10 are scatter plots of neutralizing antibody levels (%FR) that compare sow to piglet, sow to milk, and piglet to milk, respectively. Correlation coefficients and *p*-values for each comparison on each farm appear underneath each plot and are summarized in Table 1. Lines within each plot at x = 85 and y = 85 represent the positive/negative cutoff for the HTNT assay, making four quadrants within each plot. Quadrants I and III indicate values that agree, either both results being positive or both being negative. Quadrants II and IV indicate values that disagree, with one result being positive while the other is negative.

Table 2 compares pre-farrow and post-farrow serum %FR for all sows by farm. For both farms, there is a statistically significant difference between mean pre-farrow and post-farrow antibody levels, with post-farrow having higher mean circulating antibody compared to pre-farrow. Based on a cutoff for the HTNT assay of 85%FR, Farm 1 would be classified as positive, and Farm 2 would be classified as negative for neutralizing antibodies in sow serum. For Farm 1, average piglet antibody levels (%FR) for farrowing 1 (*n* = 292) and farrowing 2 (*n*= 173) were 45.2% and 61.4%, respectively. Average milk antibody levels for Farm 1 for farrowing 1 (*n* = 90) and farrowing 2 (*n* = 58) were 92% and 84.5%, respectively. For Farm 2, average piglet antibody levels for farrowing 1 (*n* = 444) and farrowing 2 (*n* = 253) were 38.7% and 40.6%, respectively. Average milk antibody levels for Farm 2 for farrowing 1 (*n* = 141) and farrowing 2 (*n* = 79) were 77.4% and 52.5%, respectively.

## 4. Discussion

For Farm 1, there were six cohorts, with 120 females enrolled in the study, and for Farm 2, nine cohorts with 180 females were enrolled. The maximum number of females sampled was 99 for Farm 1 and 129 for Farm 2. There are various reasons for the attrition of females sampled, including culling, sow mortality, and the inability to collect samples from individual females.

Neutralizing antibodies are important for protection against viral infections, such as PEDV, and have various mechanisms for neutralizing viruses, including blocking virus attachment, aggregation of virions, reduction of internalization by endocytosis, and inhibition of metabolic events critical for viral replication [12]. The goals of this study were to determine if sow serum antibody levels are correlated with piglet serum antibody levels, if sow serum antibody levels are correlated with milk antibody levels, and if piglet serum antibody levels are correlated with milk antibody levels. If antibody levels are positively correlated, then monitoring serum antibody levels could help determine herd immune status and risk of future PEDV outbreaks. Figure 5, Figure 6, Figure 7, Figure 8, Figure 9 and Figure 10 show that while these values are weakly correlated, all correlation coefficients are less than 0.57 across all comparisons. This is partially explained by the wide variation in %FR. Thus, there are other factors influencing the antibody levels that were not evaluated in this study, such as piglet suckling order, energy status of the sow, or timing of sample collection.

The HTNT assay detects virus-neutralizing antibody isotypes, including IgG, IgM, and IgA found in serum and milk at varying concentrations. The lack of statistical significance for the comparison of pre-farrowing sow and milk antibody levels for Farm 1 may be explained by the development of lactogenic immunity. Colostrum is developed by pulling antibodies (IgG) from the circulating pool of antibodies of the sow. Colostrum is comprised of IgG, IgA, and IgM, with IgG being the predominant isotype [13]. Following ingestion of colostrum by the pig, antibodies cross the gut wall and enter the piglet’s circulation. Lactogenic immunity is developed locally in the mammary gland and is primarily comprised of IgA [13]. Since the gut wall closes (i.e., the pig can no longer absorb IgG) approximately 24 h after birth, and the pig can no longer absorb antibodies from the dam, lactogenic antibodies remain within the lumen of the gut. Although the specific antibody isotypes were not determined in this study, serum-neutralizing antibodies of piglets are likely to detect IgG, and sow milk samples are likely to detect IgA. When looking at the correlation of milk and serum samples in this study, we are likely comparing these neutralizing isotypes.

Farm 1 found no correlation between sow pre-farrowing sow serum antibody levels and milk. Additionally, no correlation was found between piglet serum and milk antibody levels following the first farrowing. A majority of milk samples were positive for neutralizing antibodies, while a majority of piglets were classified as negative. This can be explained by the timing of sample collection, as milk samples were collected closer to the time of weaning, and colostrum was not evaluated in this study. Furthermore, we did not quantify the volume of colostrum consumed and the impact on the detection of PEDV circulating neutralizing antibodies.

There is an abundance of evidence describing the gut-mammary gland-secretory IgA axis and the development of lactogenic immunity in the sow. Table 2 shows that pre-farrowing mean sow serum antibody levels are lower than post-farrowing levels. It is reasonable that sow circulating neutralizing antibody levels for PEDV would be lower pre-farrowing as she primes her mammary gland for colostrum, and the data in Table 2 support this. However, there are no reports to the authors knowledge of antibodies being developed in the mammary gland and reentering the sow circulation that could help explain the increase in mean circulating antibody levels post-farrowing.

Anecdotes from practitioners suggest approximately 6 months of protection following a PEDV outbreak based on field reports. The results of this study clearly show a decrease in circulating serum-neutralizing antibody levels between 4–8 months post-outbreak. To the authors’ knowledge, this is the first report corroborating these reports. Further support comes when considering annual replacement rates of sows. As reported by Friendship et al. [14], annual replacement rates range from 30–50% in sow farms. Thus, six months after the outbreak, a quarter of the females on the farm at the time of the outbreak would have been replaced.

During this study, clinical PEDV did not recur on these farms. Samples were taken at each farm when there was an increase in clinical signs (i.e., diarrhea) in piglets, but PEDV viral RNA was never detected during the study period in either farm. An outbreak during the study period might have shown what level of neutralizing antibody might predict protection for the herd. Previous unpublished data by Brown et al. shows that %FR > 85 in sow serum can be induced by both infection with PEDV and vaccination. However, these results also showed that PEDV could still be shed in the presence of positive HTNT results. This further demonstrates the necessity of IgA antibodies in the response to PEDV infection.

This study demonstrated a clear decline in circulating neutralizing antibody levels at the herd level around six months following an outbreak. Therefore, practitioners should focus on intervention strategies during this time, such as natural planned exposure (NPE), also known as feedback, in combination with vaccination, to bolster immune levels if the goal is to not return the herd to a naïve status. Practitioners may also focus on heightened biosecurity during this time to mitigate the entrance of PEDV into a herd. Additionally, further research is needed to develop efficacious sow vaccines that will produce a protective immunity for PEDV.

## Figures and Tables

**Figure 1 viruses-16-00324-f001:**
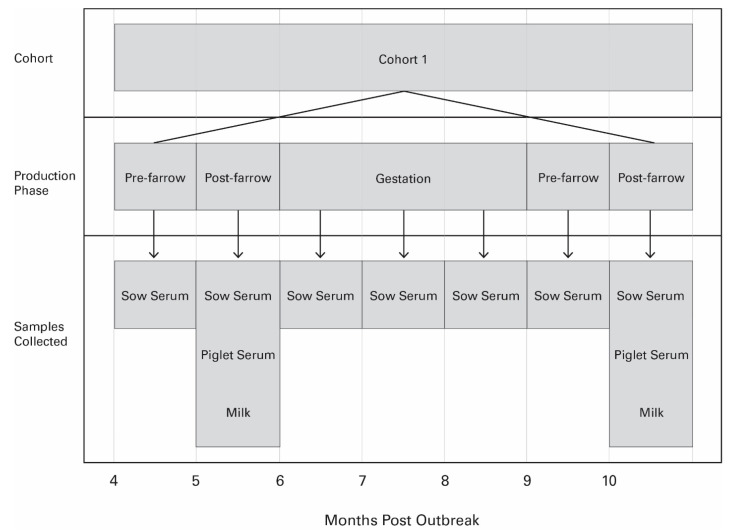
Displays the timeline of sample collection for the cohort. In this scenario, a cohort was enrolled four months post-PEDV outbreak. There were seven sample collection points, and samples were collected pre- and post-farrowing for the first farrowing, during the subsequent gestation x 3, and pre- and post-farrowing for the second farrowing. The samples collected during each production phase are outlined.

**Figure 2 viruses-16-00324-f002:**
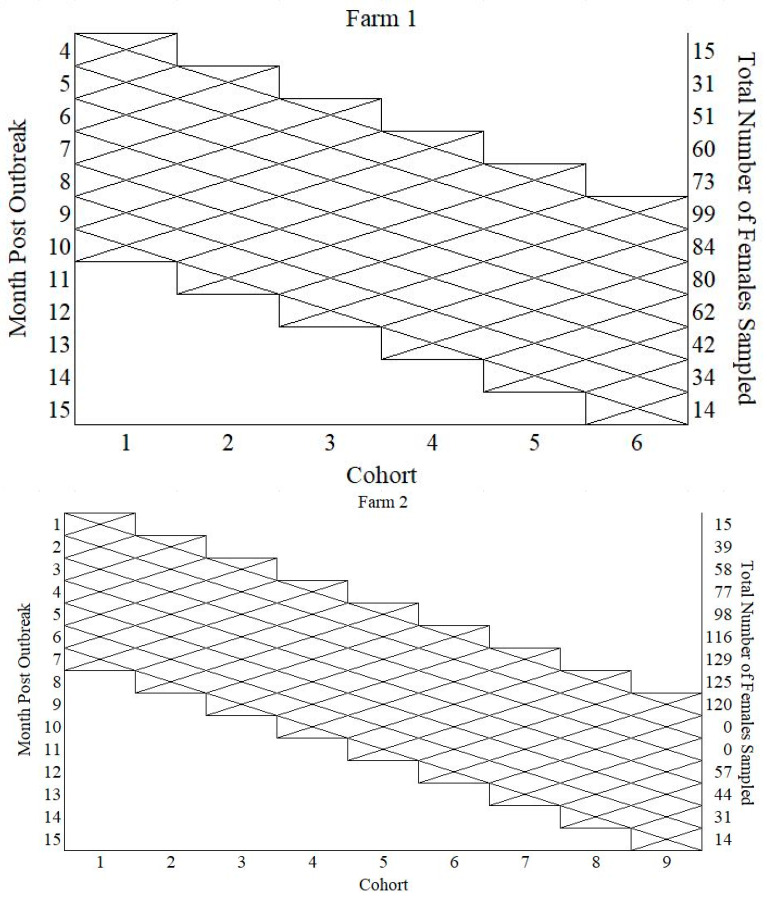
Display the total number of females sampled by month post-outbreak for Farm 1 and 2 (i.e., 99 females were sampled in month 9 post-outbreak for Farm 1).

**Figure 3 viruses-16-00324-f003:**
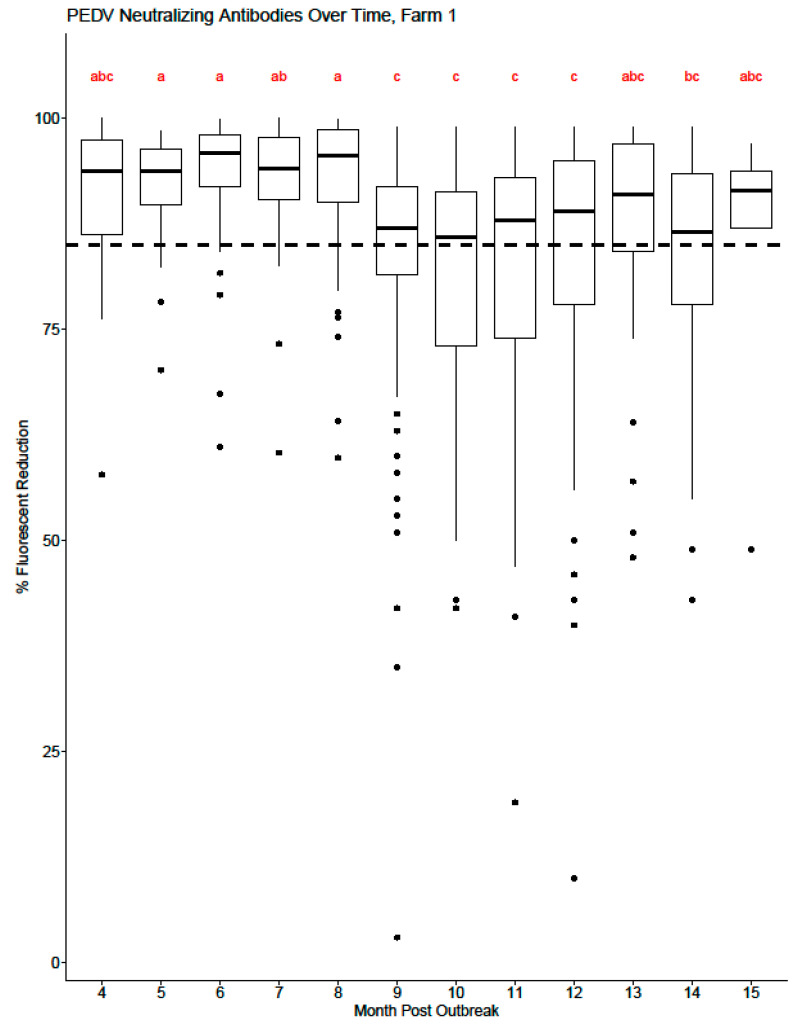
Box and whisker plot of PEDV serum-neutralizing antibody levels from Farm 1 by month post-outbreak. The dashed line at y = 85 represents the cutoff value for the HTNT assay: samples ≥ 85 are considered positive, while samples less than 85 are considered negative. The shaded box represents the 25th percentile (Q_1_), median, and 75th percentile (Q_3_). The lower whisker represents Q_1_ − 1.5 * interquartile range (IQR). The upper whisker represented the Q_3_ + 1.5 * IQR. The dots represent potential outliers. The values that share superscripts are not significantly different from each other.

**Figure 4 viruses-16-00324-f004:**
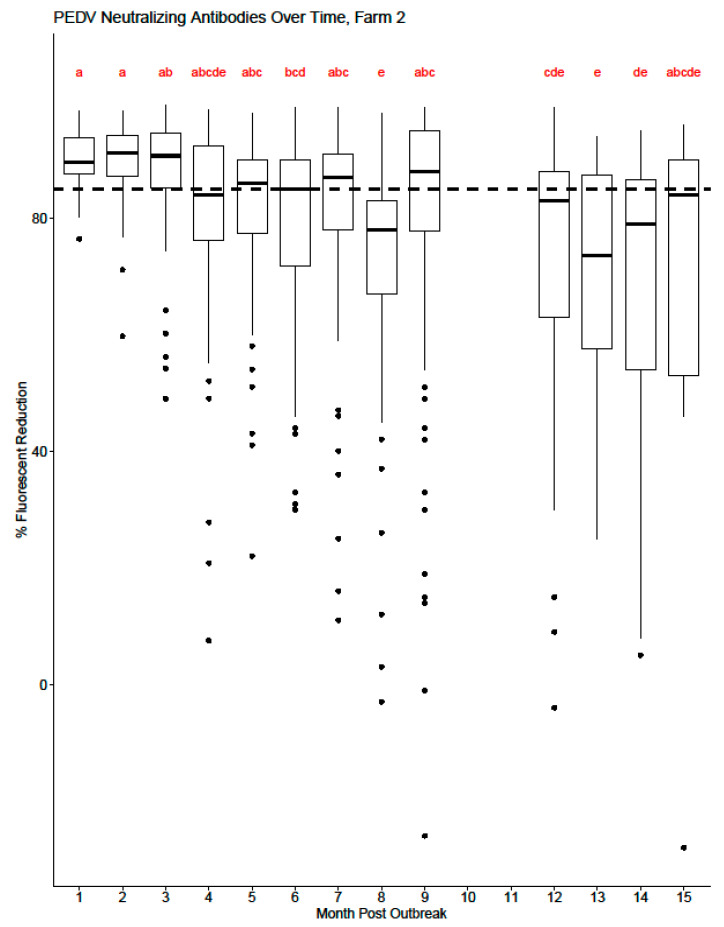
Box and whisker plot of PEDV serum-neutralizing antibody levels from Farm 2 by month post-outbreak. The dashed line at y = 85 represents the cutoff value for the HTNT assay: samples ≥ 85 are considered positive, while samples less than 85 are considered negative. The shaded box represents the 25th percentile (Q_1_), median, and 75th percentile (Q_3_). The lower whisker represents Q_1_ − 1.5 * interquartile range (IQR). The upper whisker represented the Q_3_ + 1.5 * IQR. The open circles represent potential outliers. No data was collected during months 10 and 11 post-outbreak due to the inability to access the farm location. Values that share superscripts are not significantly different from each other.

**Figure 5 viruses-16-00324-f005:**
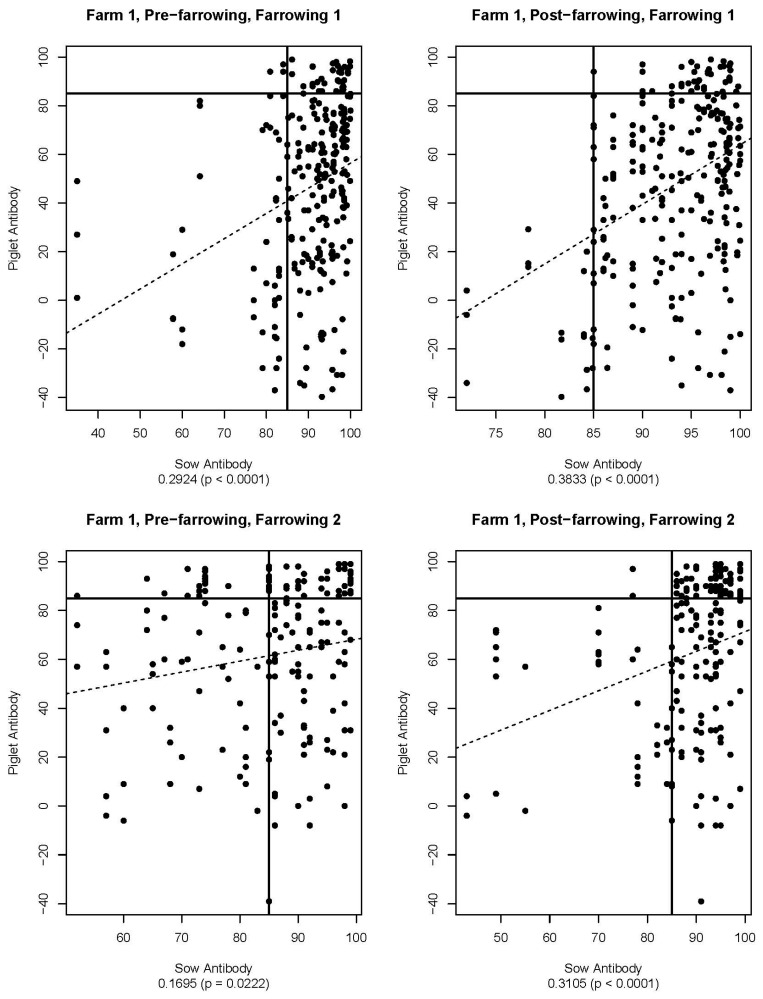
Scatter plots of sow and piglet antibody HTNT results for Farm 1. Note that *x*- and *y*-axis scales differ for each plot. The bold black lines are consistent on each plot marked at x = 85 and y = 85 as the cutoff for the HTNT assay. The dashed line represents the regression line for the data set. Below each plot, the correlation coefficient and *p*-value are listed.

**Figure 6 viruses-16-00324-f006:**
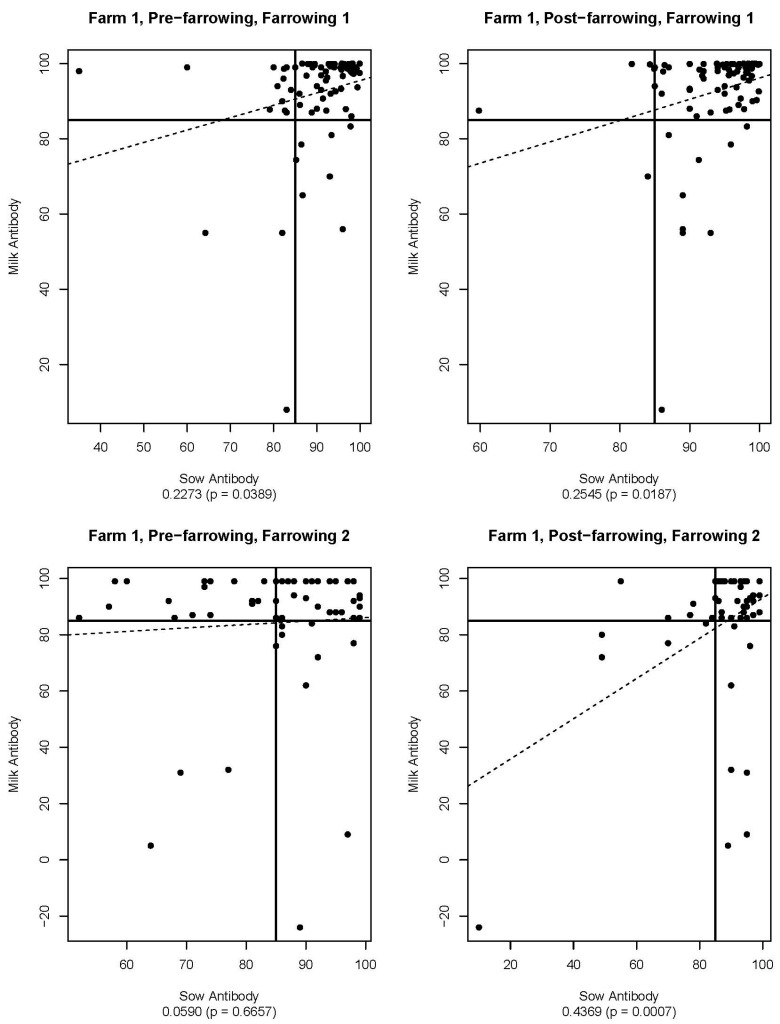
Scatter plots of sow and milk antibody HTNT results for Farm 1. Note that *x*- and *y*-axis scales differ for each plot. The bold black lines are consistent on each plot marked at x = 85 and y = 85 as the cutoff for the HTNT assay. The dashed line represents the regression line for the data set. Below each plot, the correlation coefficient and *p*-value are listed.

**Figure 7 viruses-16-00324-f007:**
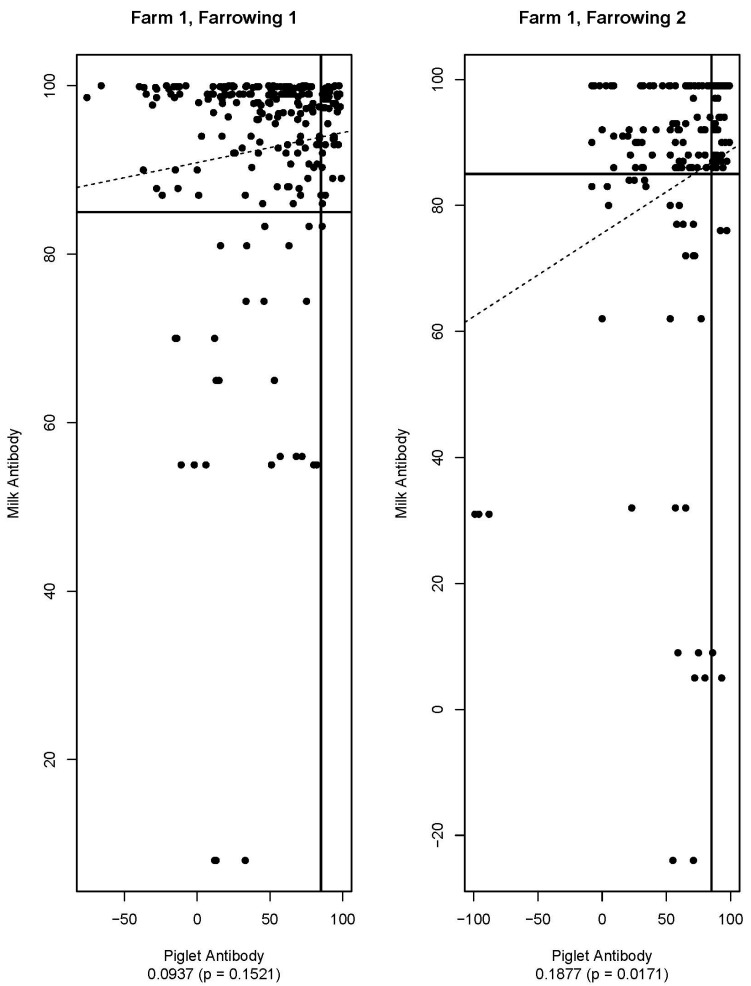
Scatter plots of piglet and milk antibody HTNT results for Farm 1. Note that *x*- and *y*-axis scales differ for each plot. The bold black lines are consistent on each plot marked at x = 85 and y = 85 as the cutoff for the HTNT assay. The dashed line represents the regression line for the data set. Below each plot, the correlation coefficient and *p*-value are listed.

**Figure 8 viruses-16-00324-f008:**
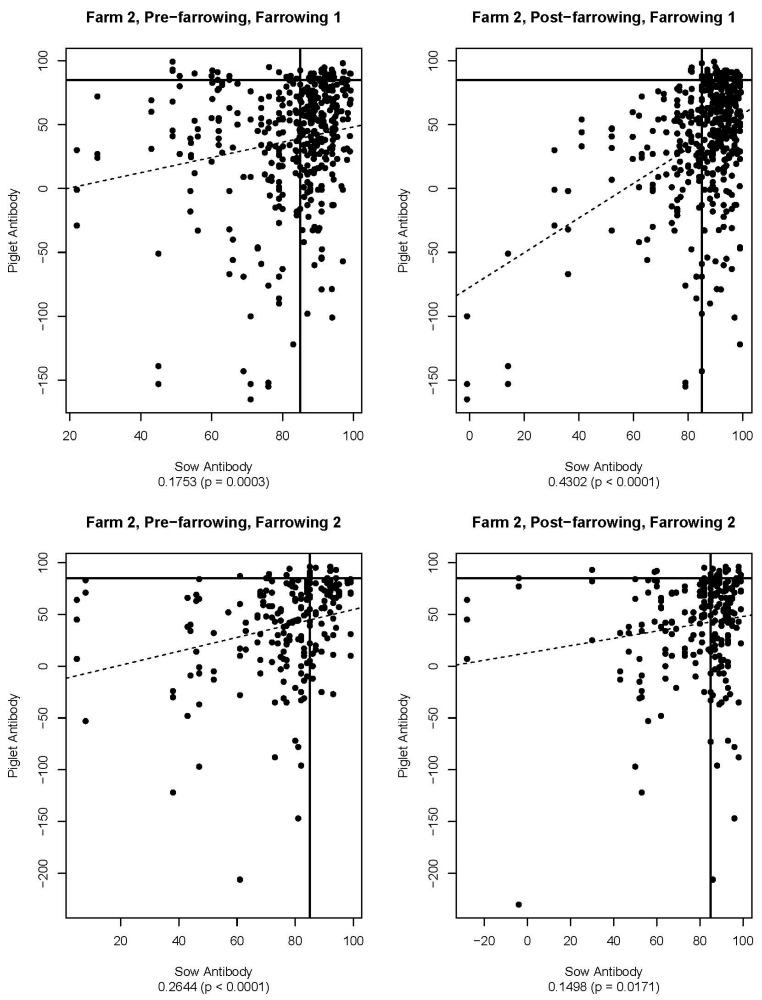
Scatter plots of sow and piglet antibody HTNT results for Farm 2. Note that the *x*- and *y*-axis scales differ for each plot. The bold black lines are consistent on each plot marked at x = 85 and y = 85 as the cutoff for the HTNT assay. The dashed line represents the regression line for the data set. Below each plot, the correlation coefficient and *p*-value are listed.

**Figure 9 viruses-16-00324-f009:**
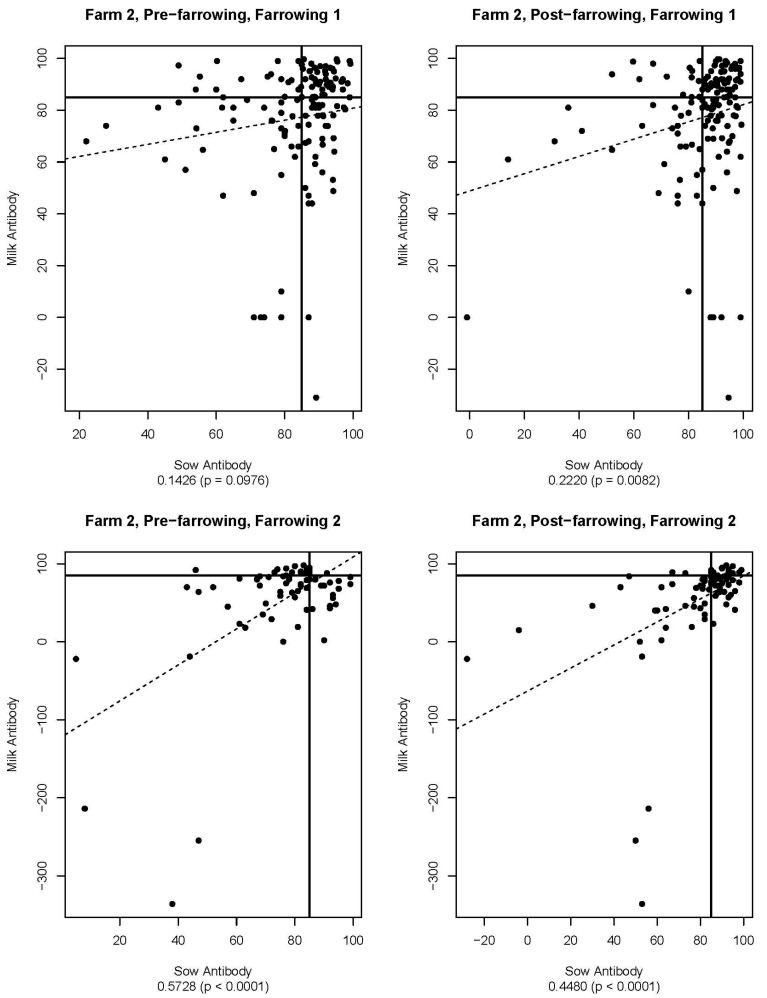
Scatter plots of sow and milk antibody HTNT results for Farm 2. Note that *x*- and *y*-axis scales differ for each plot. The bold black lines are consistent on each plot marked at x = 85 and y = 85 as the cutoff for the HTNT assay. The dashed line represents the regression line for the data set. Below each plot, the correlation coefficient and *p*-value are listed.

**Figure 10 viruses-16-00324-f010:**
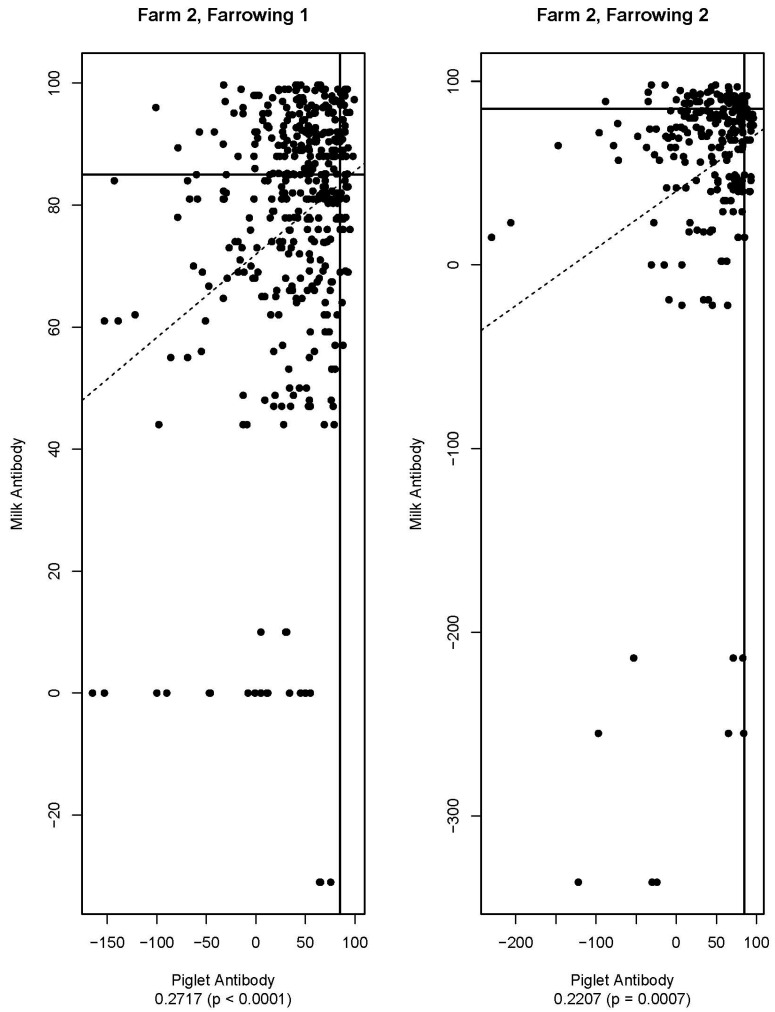
Scatter plots of piglet and milk antibody HTNT results for Farm 2. Note that *x*- and *y*-axis scales differ for each plot. The bold black lines are consistent on each plot marked at x = 85 and y = 85 as the cutoff for the HTNT assay. The dashed line represents the regression line for the data set. Below each plot, the correlation coefficient and *p*-value are listed.

**Table 1 viruses-16-00324-t001:** Correlation coefficients for sample comparisons by farm.

		Pre-Farrow 1	Farrowing 1	Post-Farrowing 1	Pre-Farrow 2	Farrowing 2	Post-Farrowing 2
	Sample	Sow	Milk	Sow	Sow	Milk	Sow
Farm 1	Piglet	0.2924 *	0.0937 *	0.3833 *	0.1695 *	0.1877 *	0.3105 *
	Milk	0.2273 *		0.2545 *	0.0590		0.4369 *
Farm 2	Piglet	0.1753 *	0.2717 *	0.4302 *	0.2644 *	0.2207 *	0.1498 *
	Milk	0.1426		0.2220 *	0.5728 *		0.448 *

* Indicates statistical significance *p* < 0.05.

**Table 2 viruses-16-00324-t002:** Comparison of pre-farrowing vs. post-farrowing sow serum %FR.

	Farm 1				Farm 2			
	mean	SE	df	95% CI	mean	SE	df	95% CI
Pre-farrow	86.9	0.857	118	(85.2, 88.6)	78.9	1.21	177	(76.5, 81.3)
Post-farrow	90.1 *	1.022	118	(88.0, 92.1)	82.3 *	1.26	177	(79.8, 84.8)

* Indicates a statistically significant difference between pre- and post-farrow mean antibody levels.

## Data Availability

The data presented in this study are available on request from the corresponding author on reasonable request.

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
