# Peer review of "Longitudinal and Cross-Sectional Evaluation of Two Commercial Swine Breeding Herds to Characterize Neutralizing Antibody Levels following Porcine Epidemic Diarrhea Virus Outbreaks"

_viruses, 2024, doi:10.3390/v16030324_

Round 1

Reviewer 1 Report

Comments and Suggestions for Authors

In this manuscript “Longitudinal and Cross-sectional Evaluation of Two Commercial Swine Breeding Herds to Characterize Neutralizing Antibody Levels Following Porcine Epidemic Diarrhea Virus Outbreaks” by Brown et al., the authors have evaluated neutralization antibody durability in sows and piglets followed through farrowing post-PEDV outbreak. This study is of importance in understanding the protective antibody response against PEDV, and enable development of better intervention strategies.

Specific comments:

1.       Include details on the year and month of sample collection in the method section. This will be enable the reader to understand the outbreak timeline - is it recent outbreak, or a retrospective study using samples from past outbreak?

2.       Introduction: L.No.34-37 - can be removed. Listing all the authors’ names of the cited reference is unnecessary.

3.       Result section needs to be restructured - make it concise and precise description of the experimental results. List the figure/table in parenthesis, in the current format it is very hard to understand.

4.       Fig.2 and 3 can be combined into a table. The figure is hard to understand.

5.       Fig.4 and 5 can be combined into one fig, and list only the statistically significance in superscript, non-significance can be represented as N.S. Too many super-script is making the data to miss its impact.

6.       Nt test: What additional dilutions were tested? Or is this a qualitative assay to measure foci reduction in 1:40 dilution? If so, it needs to be clearly mentioned in the text. “Antibody levels” usually refers to titers i.e. NT50.

7.       Fig6 to 11 - y-axis - scale has negative value. What does -100 piglet Ab mean?

8.       The data from Fig.6 to 11 can be condensed into a table. Correlation between Nt Ab titers in various samples can be just mentioned in the text.

9.       Discussion: Can be condensed, several lines are redundant. Without determining antibody isotype profile in the samples, it would be unwarranted to over speculate on this regard (L.No.240-253).

Reviewer 2 Report

Comments and Suggestions for Authors

In this manuscript, Justin Brown et al. investigate PEDV outbreaks in two commercial breeding herds in northwest Iowa. The research examines neutralizing antibody levels against PEDV in pigs. The key focus is on correlating sow pre-farrowing serum antibodies with milk antibodies, and piglet serum antibodies with milk antibodies. They reported a notable decrease in neutralizing antibodies around six months post-outbreak at the herd level.

1. In the introduction, it is necessary to provide an introduction to the current epidemic situation of PED in the United States and globally.

2. It is necessary to indicate when this work was carried out.

3. In addition to scatter plots of neutralizing antibody levels (%FR) that compare sow to piglet, sow to milk, and piglet to milk respectively, it is also essential to display the mean levels of neutralizing antibodies expressed by sow, piglet, and milk.

4. Samples from uninfected animals are required as controls.

Comments on the Quality of English Language

Many typos need to be corrected. Some examples:

Line 102, incubated at 37°C with 5% CO2 for 5 days. --> incubated at 37°C with 5% CO2 for 5 days.

Line 104, TCID50 --> TCID50

Line 109, 96-well plate (Coring 96 well CellBind microplate, Sigma) --> 96-well plate (Corning 96 well CellBind microplate, Sigma)

Line 129, using 30 ms exposure time and 20 um focal adjustments --> using 30 ms exposure time and 20 μm focal adjustments
